# Dynamic Amplitude-Frequency Characteristics of Vertical–Torsional Coupling System with Harmonic Response in Hot Tandem Mill

**Lidong Wang, Xiaoqiang Yan \*, Xingdou Jia and Xiaoling Wang**

School of Mechanical Engineering, University of Science and Technology Beijing, Beijing 100083, China
\* Correspondence: yanxq@ustb.edu.cn; Tel.: +86-186-0026-0898

**Abstract:** Vibration is a common and urgent technical issue in the steel industry. The world's first multi-mode continuous-casting and rolling plant of Shou Gang Jing Tang Iron and Steel Co., Ltd. (Tangshan, China), has a finishing mill, F3, that experiences frequent, strong vibrations during the process of rolling thin-gauge (<1.5 mm) strip steel, which have seriously hindered the production of high-quality thin strip steel products. The changes in the strips' surface quality are among the factors that induce rolling mill vibrations. In this study, considering the nonlinear surface quality of strip steel, a finite element model of the F3 mill was established, and the harmonic response method was used to obtain a rolling mill vertical system in the ANSYS environment. This study assesses the sensitive amplitude versus frequency characteristics curve of a torsional coupling system, the influence of strip thickness and strip hardness fluctuations on the vibration of the primary drive system, and the dynamic amplitude versus frequency characteristics of the three directions on the top of a torii. Finally, the field experiment verifies the correctness of the analytical results, which provides theoretical guidance for suppressing rolling mill vibrations and has a certain application value.

**Keywords:** hot tandem mill; harmonic response; coupling system; vibration

## 1. Introduction

The vibration of a strip mill directly affects the quality of the rolled products. When the vibration is severe, it causes strip breakage, stacking, or equipment damage, resulting in significant economic losses. The suppression of vibrations has become a worldwide issue that interests many scholars. At the Shou Gang Jing Tang Iron and Steel Company's first multi-mode continuous-casting and rolling plant (MCCR), the frequent strong vibrations not only affect the stable operation of the equipment but also hinders the improvement of high-quality thin strip products, as shown in Figure 1.

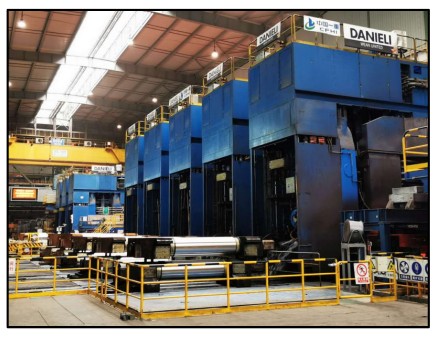
(**a**)

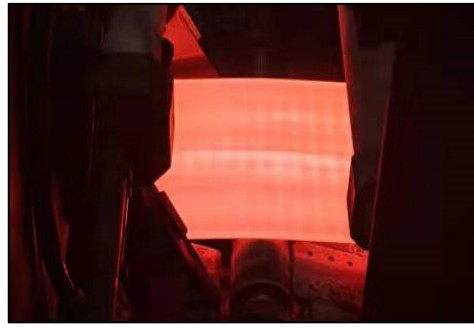
(**b**)

**Figure 1.** Tandem mill: (**a**) MCCR finishing mills; (**b**) Strip vibration chart.

In the past few decades, the vibration mechanism of rolling mills has been extensively researched. Peng et al. [1] revealed that the torque of the second finishing mill was greatly affected by vibration and considered that the vibration mode of the work roll system was generally a horizontal inversion and vertical in phase. In ref. [2], A dynamic rolling force model for predicting slab thickness fluctuations during rolling based on machine learning objectives was established. Younes et al. [3] utilized the key process empirical models to predict the flutter energy and improve the strips' surface quality by analyzing the slab temperature, exit velocity, and reduction parameters; consequently, product rejections dropped by 61%. The work roll consumption, rolling force, and reduction ratio of an F2 stand decreased by 15%, 15%, and 8%, respectively. In ref. [4], a nonlinear coupled dynamics model, based on Hurwitz's algebraic criteria, was built to investigate the effects of dynamic parameters on the horizontal, vertical, and torsional vibration modes. Tension fluctuation has a great influence on the vertical and torsional vibration of the rolling mill, while the horizontal vibration and vertical vibration stability are greatly affected by the thickness of the strips' entrance. Peng et al. [5] utilized the nonlinear singular theory and averaging method to study the dynamic amplitude versus the frequency characteristics and static bifurcation. It was found that the coupled vibration behavior is relatively complex under different bifurcation behaviors.

Qian et al. [6] proposed a novel adaptive, active fuzzy vibration controller, and the verification results showed that it can effectively suppress flutter. In ref. [7], the torsional vibration and nonlinear bifurcation vibration stability domain of the transmission system was investigated. The existence of supercritical bifurcation behavior in the coupled transmission system of the rolling mill was found by Peng et al. [8], who investigated the primary driveline stability effects of nonlinear shaft offsets. It was found that the periodic vibration energy of the rolling mill is due to the eccentricity of the transmission system. Lu et al. [9] utilized the nonlinear time-varying theory to study the dynamic coupling flutter of a rolling system. The results showed that the effective method to suppress the vibration was to adjust the reduction and optimize the lubrication. In ref. [10], the nonlinear dynamic behavior of the global bifurcation properties was studied. Studies have shown that the strip thickness fluctuation is one of the factors that lead to changes in the rolling force.

In recent years, many researchers have focused on vibration suppression and the vibration mechanism and have observed a few significant results. Moreover, such vibrations have been found to occur because of the change of the natural amplitude versus frequency characteristics. However, few researchers have focused on the influencing factors of the natural amplitude versus frequency characteristics of a finishing mill. In this study, the sensitive amplitude versus frequency characteristics of the primary drive and the vertical systems in the vertical–torsional coupling state of a hot tandem mill F3 unit are first studied by the harmonic response analysis method in the ANSYS environment. Thereafter, the influence of the strip thickness and hardness fluctuations on the dynamic amplitude versus frequency characteristics of the finishing mill in question is studied. Finally, the variation law of the finishing mill's dynamic amplitude versus frequency characteristics is verified by experiments. The results show that the impact of the strip thickness and hardness fluctuation on the rolling mill vibration is stronger than that of the influences of the thickness fluctuation. The findings of this study may be of interest to the industry.

## 2. Mathematical–Physical Model and FE Model

### 2.1. Brief Introduction of the F3 Rolling Mill

The three-dimensional geometry of the F3 frame is shown in Figure 2. The F3 frame primarily consists of a motor, gear coupling, gearbox gear, gear seat gear, gear spindle, mill stand, back-up roll, work roll, and crossbeam. The role of the motor is to provide power to the primary drive system; the function of the gearbox gear is to adjust the speed of the motor to the speed required by the roll. The role of the gear seat gear is to transmit the torque and speed of the motor to the upper and lower rolls. The motor and roll are connected by a shaft to transmit speed and torque. The work roll is the working part of the

finishing mill to complete the rolling process. The backup rolls are in contact with the work roll, and their primary function is to enhance the rigidity and strength of the work roll. The mill stand is used to install all the parts in the work stand, such as the rolls, roll bearings, and roll adjustment devices, and bear the effect of all the rolling forces. In this study, the MCCR line-finishing mill's F3 mill was selected. The primary parameters of the F3 frame are shown in Table 1.

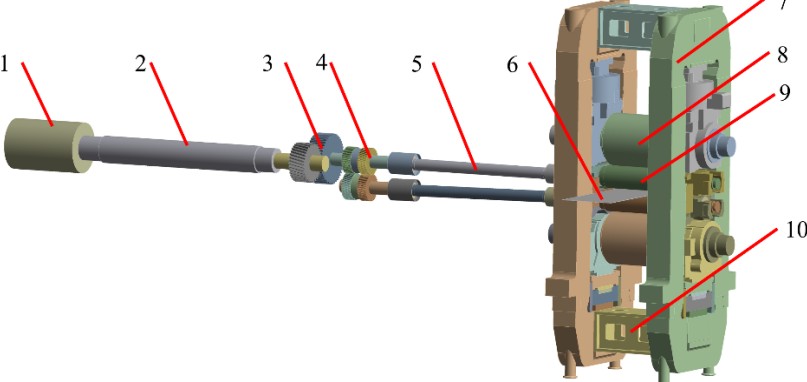

**Figure 2.** Three-dimensional geometry of the F3 rolling mill: 1—motor; 2—gear coupling; 3—gearbox gear; 4—gear seat gear; 5—gear spindle; 6—strip; 7—mill stand; 8—backup roll; 9—work roll; 10—crossbeam.

**Table 1.** Main parameters of F3 frame.

| Technical Name | Specification |
|---|---|
| Backup roll diameter | 1450 mm |
| Work roll diameter | 640 mm |
| Motor rotor moment of inertia | 6217 Kg·m$^2$ |
| Reduction ratio | 1:1.3 |
| Reducer's moment of inertia | 1370 Kg·m$^2$ |
| The number of teeth of the reducer | 36/47 |
| Gear seat's number of teeth | 47 |
| Gear seat's moment of inertia | 2350 Kg·m$^2$ |
| Poisson's ratio | 0.3 |
| Young's modulus | $2.1 \times 10^{11}$ GPa |
| Density | 7800 |
| Strip steel grades | Q235b |

The finite element model of the F3 frame was created based on the drawings, data, and hexahedral meshing, as shown in Figure 3.

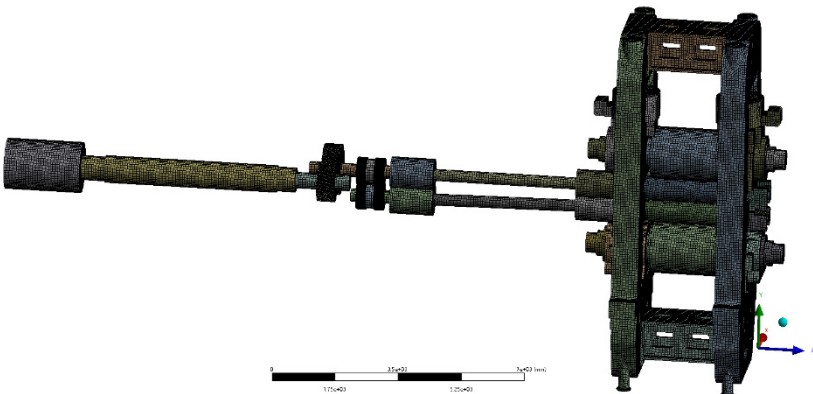

**Figure 3.** F3 frame model.

### 2.2. Mathematical and Physical Model of Vertical–Torsional Coupling

The rolling process was completed by the rolls, which extruded the strip to produce plastic deformation. Therefore, a vertical–torsional coupling model of the finishing mill is reasonable for establishing a model that includes the strip factor, as shown in Figure 4.

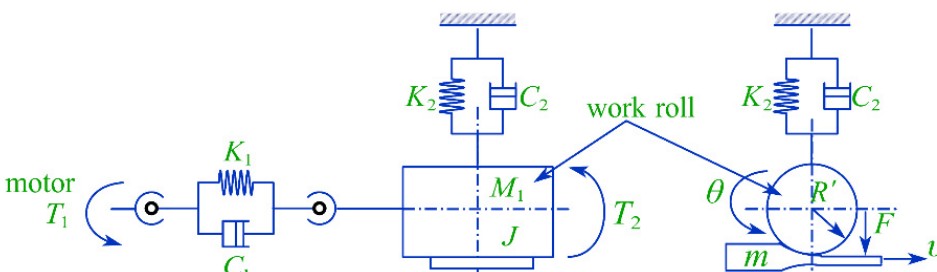

**Figure 4.** Simplified model of F3 vertical–torsional coupling.

The coupled vibration method was used to construct an equivalent virtual moment $\Delta T$ in the vertical system model, and the virtual rotation angle generated by the equivalent virtual moment acted on the vertical system. An equivalent virtual force $\Delta F$ was constructed in the model of the primary drive system, and the equivalent virtual force generates the virtual displacement acting on the primary drive system.

$$\begin{cases} \Delta F = K_r \Delta\varphi \\ \Delta T = K_r \Delta x \end{cases} \tag{1}$$

$\Delta\varphi$ is the relative angular displacement of the virtual moment in the transmission direction, $\Delta x$ is the displacement generated by the virtual force in the vertical direction, and $K_r$ is the vertical–torsional-coupling stiffness.

The state space variables of the vertical vibration are set as $y = x_2$ and $\dot{y} = x_1$, and the state space variables of torsional vibration are set as $\theta = x_4$ and $\dot{\theta} = x_3$.

Based on Figure 4, the state space expression of the vertical–torsional coupled vibration is:

$$\begin{cases} \begin{pmatrix} \dot{x}_1 \\ \dot{x}_2 \\ \dot{x}_3 \\ \dot{x}_4 \end{pmatrix} = \begin{pmatrix} -\frac{C_2}{M_1} & -\frac{K_2}{M_1} & 0 & \frac{K_r}{M_1} \\ 1 & 0 & 0 & 0 \\ 0 & \frac{K_r}{J} & -\frac{C_1}{J} & \frac{K_1}{J} \\ 0 & 0 & 1 & 0 \end{pmatrix} \begin{pmatrix} x_1 \\ x_2 \\ x_3 \\ x_4 \end{pmatrix} + \begin{pmatrix} -\frac{1}{M_1} & 0 \\ 0 & 0 \\ 0 & \frac{1}{J} \\ 0 & 0 \end{pmatrix} \begin{pmatrix} F \\ T_1 - FR'\mu \end{pmatrix} \\ \begin{pmatrix} y_1 \\ y_2 \end{pmatrix} = \begin{pmatrix} 0 & 1 & 0 & 0 \\ 0 & 0 & 0 & 1 \end{pmatrix} \begin{pmatrix} x_1 \\ x_2 \\ x_3 \\ x_4 \end{pmatrix} \end{cases} \tag{2}$$

where $C_1$ is the damping between the roll and the drive shaft, $J$ is the rotational inertia of the roll, $K_1$ is the stiffness between the work roll and the drive shaft, $T_1$ is the input torque of the motor, $R'$ is the flattening radius of the work roll, $\mu$ is the mixed friction coefficient, $C_2$ is the equivalent damping, $M_1$ is the equivalent mass of the upper work roll, $K_2$ is the equivalent stiffness of the work roll and the upper backup roll, and $F$ is the rolling force in the rolling lubrication zone.

### 2.3. Verification of the Developed Models

We understood that to determine the vibration state of the MCCR finishing mill, the installation of a vibration online monitoring system is necessary to monitor the vertical, axial, and horizontal vibration signals of the archway and the torsional vibration signal of the F3 motor output shaft. Figure 5 shows the test during the rolling process.

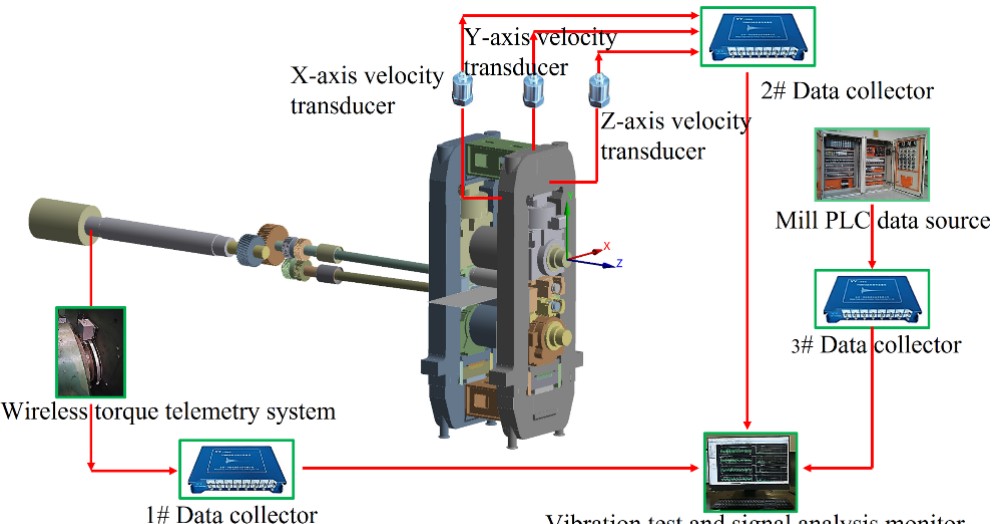

**Figure 5.** Test during rolling.

### 2.3.1. Vibration Test of Vertical System of Rolling Mill

After a long-term field test, it was found that the roof of the operating side of the rolling mill was experiencing obvious vibrations and it was convenient to install the test sensor on site. Therefore, three magnetic vibration velocity sensors (as shown in Figure 5) were installed on the top of the arch on the operating side of the F3 rolling mill to measure the vertical vibration velocity of the F3 operating side (Y-axis velocity), and the axial vibration velocity of the F3 operating side (Z-axis velocity) and the F3 operating side. Concerning the vibration velocity in the horizontal direction (X-axis velocity), when the measured point vibrates, the output signal of the vibration velocity sensor is sent to the 2# collector, vibration test, and signal analysis monitor. The software adopts the YSV dynamic signal acquisition and analysis system, independently developed by the Beijing University of Science and Technology for time domain and frequency domain analysis; at the same time, the 3# collector introduces the rolling speed signal, strip entrance hardness signal, strip thickness signal acquisition, and the display and analysis of the stand in the PLC of the factory finishing mill into the vibration test and signal analysis monitor.

The magnetoelectric vibration velocity sensor was used in the test during the rolling process, and the important parameters of the sensor are shown in Table 2.

**Table 2.** Transducer.

| Technical Name | Specification |
|---|---|
| Frequency measurement range | 0.1 Hz–12 KHz |
| Sensitivity | 10 mV/(mm·s$^{-1}$) |
| Lateral sensitivity ratio | ≤5% |
| Amplitude linearity deviation | ±2% |
| Temperature response deviation | <0.1%/°C |
| Output load resistance | ≤500 Ω |

### 2.3.2. Vibration Test of Rolling Mill Drive System

Since the type of vibration of the transmission system of the hot tandem mill is mainly torsional vibration, the wireless torque telemetry system (as shown in Figure 6) is used for testing purposes during the rolling process, and the main parameters of the wireless torque telemetry system are shown in Table 3.

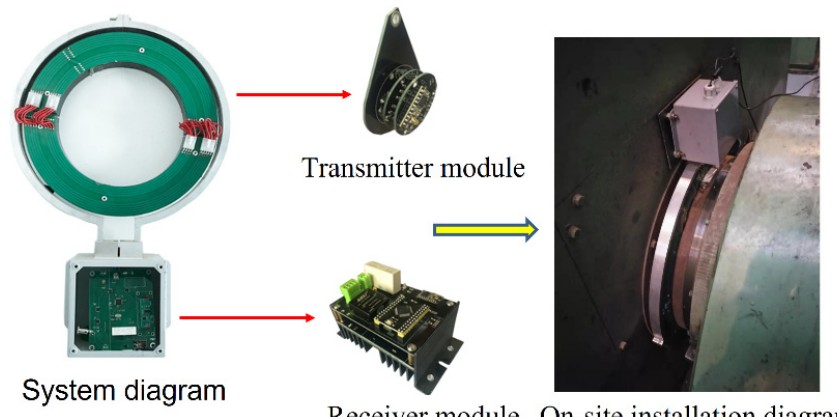

**Figure 6.** The wireless torque telemetry system.

**Table 3.** Wireless torque telemetry system parameters.

| Technical Name | Specification |
| --- | --- |
| Shaft diameter | 100 mm–1000 mm |
| Spinning speed | <9000 rpm |
| Sampling frequency | 2048 Hz/CH |
| Input range | ±2.5 V |
| Strain gauges | 100 Ω–KΩ (full bridge) |
| Strain accuracy | 0.3% |
| Power consumption | 5 V 100 mA |
| AD accuracy | 24-bit delta-sigma chip |
| Number of channels | 1 |

First, weld the torque strain gauge at a position of 45° with the axis of the output shaft of the F3 main motor under test. When the measured shaft is deformed, the torque strain gauge is deformed together with the motor output shaft, and the strain effect of the strain gauge is used to convert the deformation into the change of resistance, and finally, into the output of the voltage signal of the full-bridge circuit. The magnitude of the voltage output signal is proportional to the torque. The torque signal is modulated by the transmitter module carrier and sent to the transmitting antenna for transmission. After the receiving antenna receives the signal, it is demodulated and restored to the torque signal by the telemetry receiver module in the main control unit, and finally sent to the 1# collector and the vibration test and signal monitor. Afterwards, perform time and frequency domain analysis.

The collector was used in the test during rolling process, and the primary parameters are shown in Table 4.

**Table 4.** Collector primary parameters.

| Technical Name | Specification |
| --- | --- |
| Bandwidth | 1 Hz–10 KHz |
| Maximum sample rate | 51,200 Hz |
| Maximum input range | ±100,000 µε |
| Resolution | 0.1 µε |
| Overvoltage protection between terminals | ±60 V |
| Operating temperature | −20 to 85 °C |
| Stability | Temperature drift gain < 6 ppm/°C |
| Power supply | AC 220 V |
| Input channel accuracy | ≤0.2% |
| Input channel | 16 |
| Analog to digital converter | 24-bit AD, delta-sigma type |

## 3. Harmonic Response Analysis and Experiments

### 3.1. Sensitive Frequency Analysis

Based on the realistic rolling process of the on-site F3 finishing mill, the rolling force and moment load were applied to the work rolls' rolling surface and the motor, respectively, as shown in Figure 7:

$$F = F_0 + \Delta F \sin 2\pi f t \tag{3}$$

$$T = T_0 + \Delta T \sin 2\pi f t \tag{4}$$

where $F_0$ is a steady rolling force, $F_0$ = 20,000 KN, $\Delta F \sin 2\pi f$ is a rolling force with a fluctuating load, and $\Delta F$ = 2000 KN. The excitation frequency, $f$, is considered to be 0–200 Hz, and the harmonic response simulation is performed for every 1 Hz increment; $T_0$ is the preload torque; $T_0$ = 170 KNm; $\Delta T \sin 2\pi f t$ is the moment fluctuation load; and $\Delta T$ = 17 KNm.

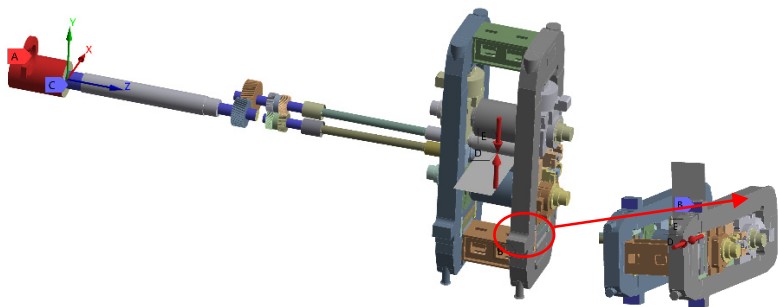

**Figure 7.** Finite element structure model: A—uniform torque; B—fixed constraints; C—cylindrical constraints; D—rolling force; E—rolling force.

Based on the field measurements, the vibration frequency of the hot tandem mill was within 200 Hz. The harmonic response simulation analysis' load excitation occurs at every 1 Hz interval.

The simulation results of the harmonic response of the output shaft's measuring point of the primary drive motor of the finishing mill is shown in Figure 8. Evident peaks were obtained at 18 and 40 Hz, indicating that the finishing mill was very sensitive to these frequencies. Furthermore, if the external excitations were close to 18 and 40 Hz, the rolling mill would experience greater torsional vibration, as shown in Figure 9. Additionally, the harmonic response of the vertical measuring point on the top of the archway had evident peaks at 64 and 121 Hz, indicating that the rolling mill was very sensitive to frequencies of 64 and 121 Hz. When the 121 Hz excitation occurred, a large vibration of the finishing mill occurred in the vertical direction.

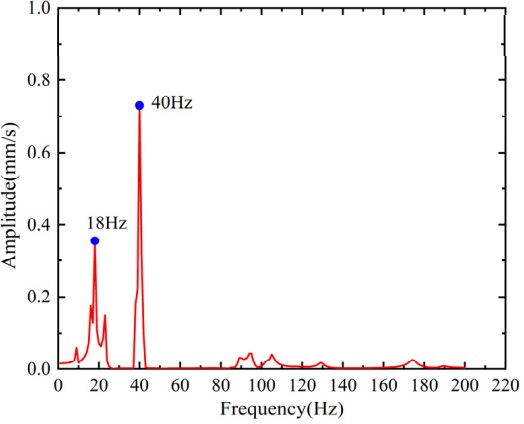

**Figure 8.** Torsional vibration characteristic curve.

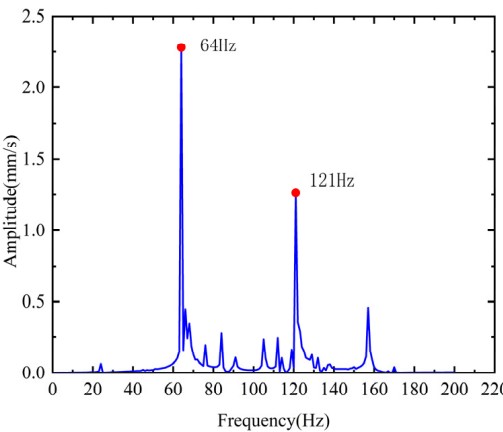

**Figure 9.** Vertical vibration characteristic curve.

As shown in Figures 10 and 11, the harmonic response simulation results of the vibration's measuring points in the axial and horizontal directions on the top of the rolling mill arch showed that three sensitive frequencies, namely, 40, 56, and 92 Hz, existed in the axial direction, and two sensitive frequencies, 40 and 70 Hz, existed in the horizontal direction, indicating that the axial and horizontal vibrations of the primary drive system of the F3 mill and the vertical system experienced a vibration-coupling phenomenon.

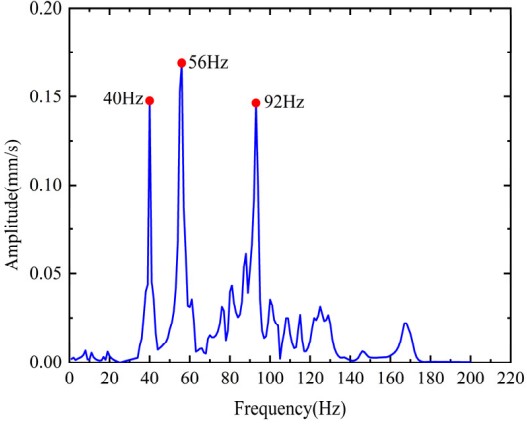

**Figure 10.** Axial vibration characteristic curve.

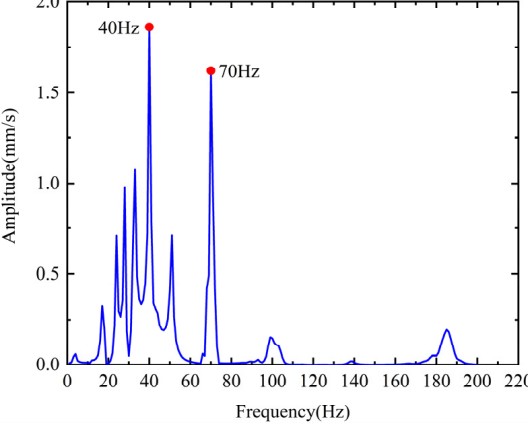

**Figure 11.** Horizontal vibration characteristic curve.

*3.2. Sensitive Frequency Variation under Strip Thickness Fluctuations*

Based on the dynamic rolling force model [10–13] and Figure 12, the relationship between the dynamic rolling force and thickness fluctuation can be represented as

$$P(X, H) = P(x_0, H_0) + \Delta P(x, \Delta H) \tag{5}$$

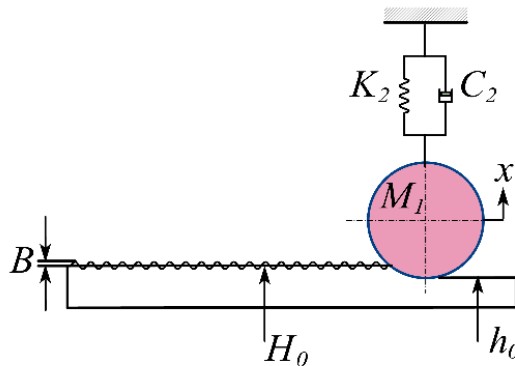

**Figure 12.** Vibration mechanics model with strip thickness fluctuation.

The dynamic rolling force consists of a steady-state rolling force and rolling force fluctuations.

$$
\begin{aligned}
\Delta P(x, \Delta H) = \ & B_1 x + B_2 \Delta H + B_3 x^2 + B_4 x \Delta H + B_5 \Delta H^2 + \\
& B_6 x^3 + B_7 x^2 \Delta H + B_8 x \Delta H^2 + B_9 \Delta H^3 + \\
& o\left( \sum_{n=0}^{3} x^n \Delta H^{3-n} \right)
\end{aligned} \tag{6}
$$

Here, $\Delta H = B\cos(\omega t)$ is the strip inlet thickness fluctuation expression. Substituting Equation (6) into Equation (5), the dynamic rolling force is expressed as $\omega$

$$
\begin{aligned}
P(X, H) = \ & P(x_0, H_0) + B_1 x + B_2 B \cos(\omega t) + \\
& B_3 x^2 + B_4 B \cos(\omega t)x + B_5 B^2 \cos^2(\omega t) + \\
& B_6 x^3 + B_7 B \cos(\omega t)x^2 + B_8 B^2 \cos^2(\omega t)x + \\
& B_9 B^3 \cos^3(\omega t)
\end{aligned} \tag{7}
$$

Figure 13 shows the influence of the strip thickness fluctuation on the torsional vibration of the primary drive system. We observed that the simulation results and field test results had the same change trend. The larger the strip thickness fluctuation at the rolling mill entrance, the smaller the sensitive frequency amplitude in the torsional vibration. However, as shown in Figure 14, with the increase in the thickness fluctuation of the strip at the entrance, the vertical vibration in the vertical system becomes increasingly evident, and the vibration amplitude is the largest among the vibrations in the three directions. The results showed that although there were a few slight differences between the experimental method and the finite element analysis, this might have been because of the modal superposition error adopted in the finite element analysis method [14,15].

As shown in Figures 15 and 16, as the fluctuation in the strip thickness increases, the amplitude of the sensitive frequency in the axial and horizontal vibrations of the vertical system also increases. However, the fluctuation did not affect the magnitude of the sensitive frequency, and the change in the thickness fluctuation did not affect the vertical vibration. The vibration in the vertical and horizontal directions had a great influence and slightly affected the axial vibration.

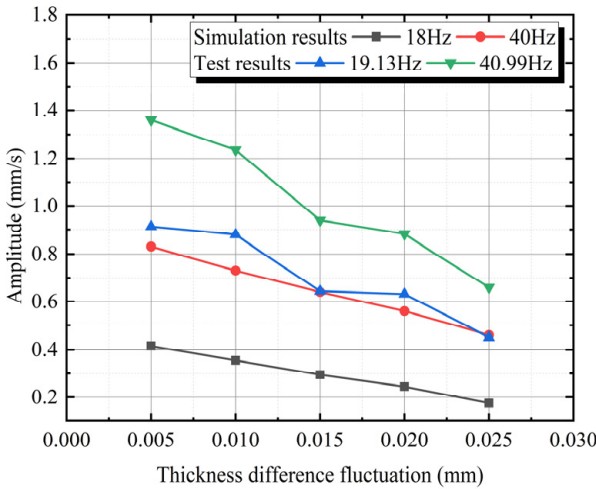

**Figure 13.** Plate thickness affects torsional vibration.

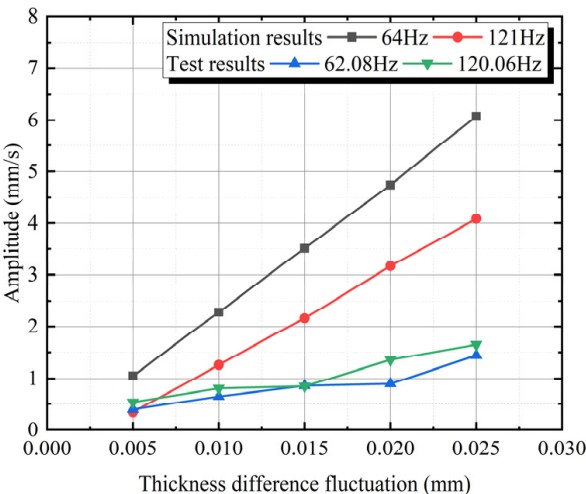

**Figure 14.** Plate thickness affects vertical vibration.

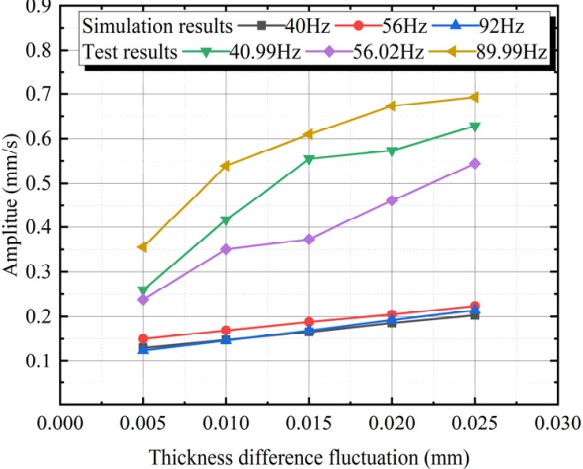

**Figure 15.** Plate thickness affects axial vibration.

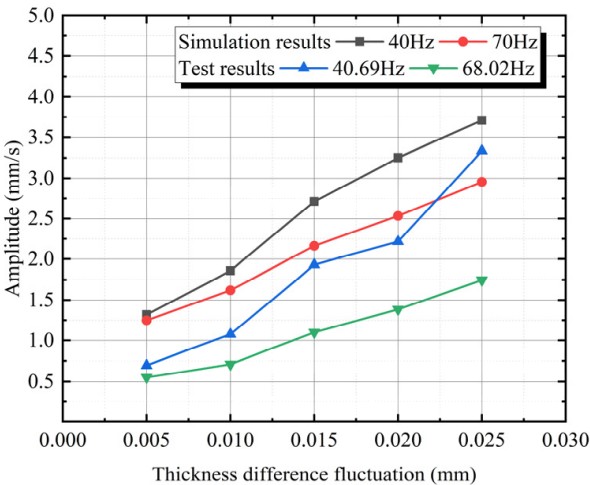

**Figure 16.** Plate thickness affects horizontal vibration.

### 3.3. Sensitive Frequency Changes under Strip Hardness Fluctuations

According to SIMS theory [16–19] and Figure 17, the force balance equation in the horizontal direction is listed as

$$\frac{d}{d\phi}\left(\frac{p_\phi}{k} - \frac{\pi}{4}\right) = \frac{R'\pi\phi}{2(h_c + R'\phi^2)} \pm \frac{R'}{h_c + R'\phi^2} \tag{8}$$

where $K$ is the deformation resistance and $R'$ is radius of the work rolls after deformation. Since $\phi$ is very small, it can be ignored. Therefore, the change in the rolling force per unit width is:

$$dF_R = \frac{dF_R}{dk}dk + \frac{dF_R}{dh_c}dh_c + \frac{dF_R}{d\dot{h}_c}d\dot{h}_c \tag{9}$$

where $dk$ is the hardness fluctuation, $dh_c$ is the variation of the roll gap, and $d\dot{h}_c$ is the roll gap's change rate.

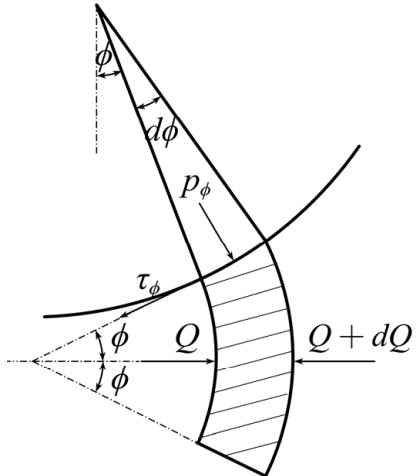

**Figure 17.** Stress distribution of the force element.

In Figure 18, the impact of the inlet strip hardness fluctuation on the torsional vibration of the primary drive system can be observed, based on the simulation and field test results.

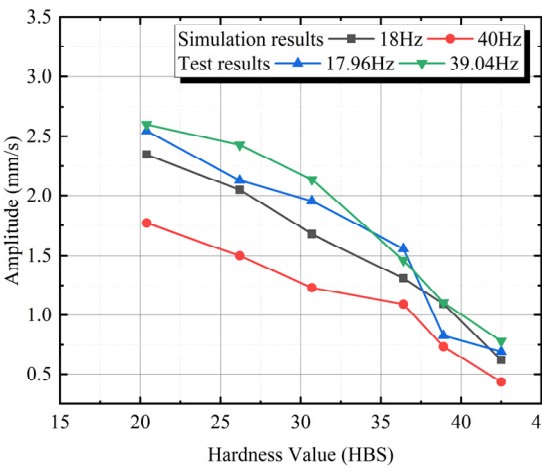

**Figure 18.** Hardness affects torsional vibration.

In Figures 19–21, it can be observed that with the increase in the strip hardness, the vertical, axial, and horizontal vibration amplitudes in the vertical system gradually decrease and the vertical and horizontal vibration decrease significantly. The fluctuation of the strip hardness had no evident effect on the axial vibration of the vertical system.

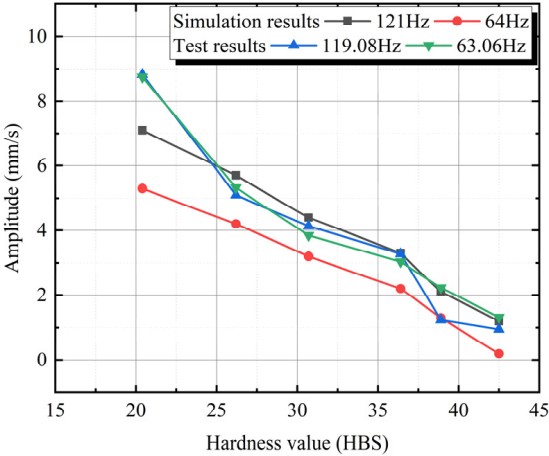

**Figure 19.** Hardness affects vertical vibration.

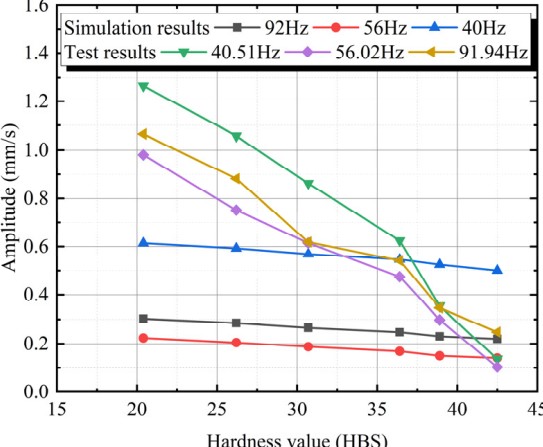

**Figure 20.** Hardness affects axial vibration.

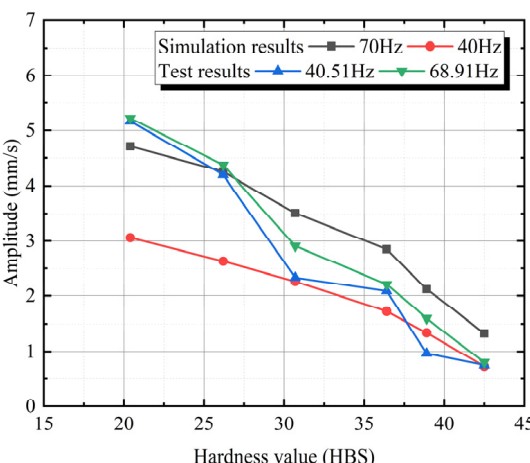

**Figure 21.** Hardness affects horizontal vibration.

### 4. Conclusions

In our current research, the entire F3 rolling mill of the MCCR finishing mill was analyzed as the main object, and the effects of the thickness and hardness fluctuations of the strip steel at the entrance of the rolling mill on the torsional, vertical, and axial vibrations were examined. The following conclusions were drawn:

(1) From the amplitude versus frequency characteristics curve of the hot tandem mill, we observed that a coupled vibration between the primary drive and vertical systems is required. Additionally, the thickness fluctuation at the entrance of the F3 rolling mill has the greatest impact on the vertical and horizontal vibrations. However, the fluctuation has less of an impact on the axial vibration, and the amplitudes of the vertical, horizontal, and axial vibrations increase with the thickness. In addition, the influence of the thickness fluctuation of the rolling mill's entrance on the torsional vibration is relatively small, and the torsional amplitude value gradually decreases with the increase in the entrance thickness fluctuation.

(2) The fluctuation of the strip hardness has the greatest impact on the vertical and horizontal vibrations of the hot tandem rolling mill and has a less of an effect on torsional and axial vibrations. The vibrational amplitude decreases gradually towards horizontal vibrations.

(3) When suppressing the vibration of the hot tandem mill, it is suggested to reduce the thickness fluctuation of the strip entry while matching the strip's quality; meanwhile, increase the strip entry hardness fluctuation.

(4) The change trend of the amplitude-frequency response curve of the hot tandem rolling mill does not increase uniformly with the fluctuation of the strip thickness, while the influence trend of the strip hardness fluctuation on the change in the amplitude-frequency response curve is uniformly decreased.

We believe there are four main reasons for the difference between the measurements and the simulation:

1. There are some necessary structural simplifications in the establishment of the F3 rolling mill stand simulation model, such that there is an error between the damping coefficient of the simulation model and the actual rolling mill stand's structural damping value.

2. There is a certain error in the setting of the restraint mode of the harmonic response analysis model in the Ansys Workbench, the addition of the load, and the actual dynamic process state of the rolling mill.

3. The sensor itself has a measurement error with respect to the field vibration signal measurement.

4. We believe that the coupling between the horizontal, vertical, and axial modes may lead to the existence of this difference, and our follow-up research will further investigate this aspect.

The results of this study have certain errors and limitations but analyzing the amplitude-frequency characteristics of natural sensitive frequencies based on a simplified model can provide theoretical references for theoretical analysis and vibrational suppression.

**Author Contributions:** Conceptualization, L.W.; methodology, L.W. and X.Y.; software, L.W.; validation, L.W., X.J. and X.W.; formal analysis, X.W.; investigation, X.J.; data curation, X.Y.; writing—original draft preparation, L.W.; writing—review and editing, L.W.; visualization, L.W.; supervision, X.Y.; project administration, X.W. and X.Y.; funding acquisition, X.Y. All authors have read and agreed to the published version of the manuscript.

**Funding:** This work is supported by the National Science and Technology Support Program of the "Twelfth Five-Year Plan", the "Precision Strip Steel Product Quality Optimization and Key Equipment Research and Development" project (2015BAF30B00), and the Fundamental Research Fund Project of Central Universities (FRF-AT-19-001).

**Conflicts of Interest:** The authors declare no conflict of interest.

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
