# Peer review of "Dynamic Amplitude-Frequency Characteristics of Vertical–Torsional Coupling System with Harmonic Response in Hot Tandem Mill"

_electronics, doi:10.3390/electronics11193031_

Round 1

Reviewer 1 Report

The paper contains good quality research work but some issues have to be solved:

1.       There are mentioned researches from the references [1] – [10] regarding the modelling and study of different vibration cases. Please write, for all of these references, which are the practical conclusions of them.

2.       According to Figures 12, 13, 14, 15, 16, 17, 18, 19  and 20, please explain how the tests have been performed (test rig, test parameters, test procedure).

3.       Please explain and show in 4 Conclusions how the results obtained in the paper fit into the same type results from the specific references. Find some references that are useful to compare the results.

4.       Please explain in 4 Conclusions which are the practical recommendations regarding the obtained results.

Reviewer 2 Report

There is a significant difference between experimental and simulation results. Is there any explanation? Does the neglected coupling between the horizontal, vertical and axial modes can cause such difference?

How the achieved results may be applied in order to suppress the vibration in the given mill plant? Does they applicable at all?

Which excitation frequency is used (line 150)?

What means ".. the simulation is performed every 1 Hz"?

Round 2

Reviewer 1 Report

Thank you, it is OK.

Author Response

Dear reviewer,

Reviewer 2 Report

The authors answered all my questions in satisfactory way. I just miss the explanation of the difference between measurement and simulation in the manuscript (it was given in the response to reviewers).

Author Response

Dear reviewer,
